# TRIM32 Deficiency Impairs the Generation of Pyramidal Neurons in Developing Cerebral Cortex

**DOI:** 10.3390/cells11030449

**Published:** 2022-01-28

**Authors:** Yan-Yun Sun, Wen-Jin Chen, Ze-Ping Huang, Gang Yang, Ming-Lei Wu, De-En Xu, Wu-Lin Yang, Yong-Chun Luo, Zhi-Cheng Xiao, Ru-Xiang Xu, Quan-Hong Ma

**Affiliations:** 1Department of Neurology and Clinical Research Center of Neurological Disease, The Second Affiliated Hospital of Soochow University, Suzhou 215123, China; yysun@suda.edu.cn (Y.-Y.S.); 20194254018@stu.suda.edu.cn (Z.-P.H.); 20204250100@stu.suda.edu.cn (M.-L.W.); 2Jiangsu Key Laboratory of Neuropsychiatric Diseases, Institute of Neuroscience, Soochow University, Suzhou 215123, China; 3Department of Neurosurgery, Sichuan Academy of Medical Sciences and Sichuan Provincial People’s Hospital, University of Electronic Science and Technology of China, Chengdu 610072, China; uscyxychenwj@126.com; 4Lab Center, Medical College of Soochow University, Suzhou 215123, China; yanggang@suda.edu.cn; 5Wuxi No. 2 People’s Hospital, Wuxi 214001, China; xudeen@njmu.edu.cn; 6Anhui Province Key Laboratory of Medical Physics and Technology, Institute of Health and Medical Technology, Hefei Institutes of Physical Science, Chinese Academy of Sciences, Hefei 230031, China; yangw@cmpt.ac.cn; 7Hefei Cancer Hospital, Chinese Academy of Sciences, Hefei 230031, China; 8Department of Neurosurgery, First Medical Center of Chinese PLA General Hospital, Beijing 100028, China; luoyong4581@plagh.cn; 9Department of Anatomy and Developmental Biology, Monash University, Clayton 3800, Australia; zhicheng.xiao@monash.edu

**Keywords:** TRIM32, excitatory-inhibitory imbalance, cortex development, ASD, NPCs

## Abstract

Excitatory-inhibitory imbalance (E/I) is a fundamental mechanism underlying autism spectrum disorders (ASD). TRIM32 is a risk gene genetically associated with ASD. The absence of TRIM32 causes impaired generation of inhibitory GABAergic interneurons, neural network hyperexcitability, and autism-like behavior in mice, emphasizing the role of TRIM32 in maintaining E/I balance, but despite the description of TRIM32 in regulating proliferation and differentiation of cultured mouse neural progenitor cells (NPCs), the role of TRIM32 in cerebral cortical development, particularly in the production of excitatory pyramidal neurons, remains unknown. The present study observed that TRIM32 deficiency resulted in decreased numbers of distinct layer-specific cortical neurons and decreased radial glial cell (RGC) and intermediate progenitor cell (IPC) pool size. We further demonstrated that TRIM32 deficiency impairs self-renewal of RGCs and IPCs as indicated by decreased proliferation and mitosis. A TRIM32 deficiency also affects or influences the formation of cortical neurons. As a result, TRIM32-deficient mice showed smaller brain size. At the molecular level, RNAseq analysis indicated reduced Notch signalling in TRIM32-deficient mice. Therefore, the present study indicates a role for TRIM32 in pyramidal neuron generation. Impaired generation of excitatory pyramidal neurons may explain the hyperexcitability observed in TRIM32-deficient mice.

## 1. Introduction

The imbalance between excitability and inhibitory activity in brain circuits is one of the key mechanisms underlying neurodevelopmental disorders, such as autism spectrum disorders (ASD), which are characterized by impaired social behaviours and repetitive and stereotypic behaviours [1]. The E/I balance in the brain is coordinated by glutamatergic pyramidal neurons and GABAergic interneurons. In cerebral cortex, the pyramidal neurons are arranged in six layers, while the inhibitory interneurons are scattered. There are definite temporal and spatial orders in which excitatory pyramidal neurons and inhibitory interneurons are generated [2,3,4]. For example, the cortical pyramidal neurons are generated following an “inside-out” order, while deep-layer cortical neurons are generated following an “out-side-in” order [5,6]. The later born cortical neurons will move across the earlier-born neurons which reside in the deep laminar of cortex [6]. Any disturbances during these sequential processes will lead to developmental disorders such as ASD.

In the developing dorsal ventricular zone, RGCs produce both upper layer and deep layer cortical pyramidal neurons. RGCs can either directly generate cortical pyramidal neurons or generate them via intermediate progenitor cells (IPCs). RGCs generate both neurons and glia while IPCs only generate neurons. Intrinsic and extrinsic signals coordinate the proliferation and differentiation of RGCs and IPCs, thus regulating the development of cerebral cortex. Exploring the molecular mechanisms underlying such regulation would help to understand the pathogenesis of developmental neurological disorders.

Tripartite motif (TRIM) 32 belongs to the TRIM family, has a RING finger-like structure as its major feature and contains one or two zinc ions called “B-Boxes” and a related coiled-coil region [7]. TRIM32 has E3 ubiquitin ligase activity [8]. It plays an important role in the ubiquitin-protease degradation of proteins. TRIM32 is expressed in a variety of systems. In the nervous system, TRIM32 is primarily expressed on neural progenitor cells (NPCs). TRIM32 is ubiquitously found in the cytoplasm of NPCs, although it translocates to the nuclei once the NPCs differentiate into neurons [9]. In cultured NPCs, knocking-down of TRIM32 increases proliferation of NPCs, while decreasing neuronal differentiation [10,11]. Rare copy number variation analysis has shown that the loss of *TRIM32* gene is strongly associated with autism and attention deficit hyperactivity disorder [12,13]. Consistent with this finding, TRIM32 knockout mice exhibit ASD-like behaviors and hyperexcitability, accompanied with decreased numbers of interneurons in the telencephalon [14]. These studies indicate that TRIM32, as an essential modulator in NPCs, play import roles in maintenance of cortical development. In this paper, we describe TRIM32’s role in the production of pyramidal neurons in developing cortex. Absence of TRIM32 leads to a smaller size of brain. TRIM32-deficient mice exhibit decreased numbers of both upper- and deep-layer cortical neurons, accompanied with a reduced proliferation of both RGCs and IPCs. According to the present study, loss of TRIM32 impairs the generation of cortical pyramidal neurons by reducing the size of the pool of NPCs.

## 2. Materials and Methods

### 2.1. Mice

TRIM32 KO mice (TRIM32^−/−^) were kindly provided by Professor Jens C. Schwamborn from the University of Luxembourg. The mouse uses the BGA355 embryonic parent cell line to capture and insert a 5 kb genomic fragment into the second exon of TRIM32 with a SA-IRES-bgeopA expression cassette. This mouse background is a 129SvEvBrd X C57 BL/6 heterozygote and then reverts to C57 BL back/6 precursors for more than 8 generations [15]. Adult mice were reared on an adequate supply of food and water for 12 h. Day-night light cycle at 25 °C. Mice used in this study were from heterozygous breeding pairs (TRIM32^+/−^) and TRIM32^+/+^ littermates were used as controls. All animal experiments were carried out following the Institutional Animal Care and Use Committee of Soochow University.

### 2.2. Genotyping Detection of TRIM32 Mice

Before using the animals in different experiments, TRIM32 mice were genotyped. We used the following primer sequences to detect genotyping: TRIM32 WT1 (5′-3′): GGAGAGACACTATTTCCTAAGTCA; TRIM32 WT2 (5′-3′): GTTCAGGTGAGAAGCTGCTGCA; TRIM32 Mu (5′-3′): GGGACAGGATAAGTATGACATCA. The primer pairs WT1 and WT2 and WT1 and Mu were used to designate WT and knockout mice, respectively. The PCR reaction conditions were set at 94 °C (5 min) for enzyme activation, 35 cycles of denaturation at 94 °C for 30 s, annealing at 60 °C for 30 s, extension at 72 °C for 30 s, then 72 °C for 7 min and finally at 4 ◦C infinity. The amplified DNA was separated on a 2% agarose gel by electrophoresis using a current of 120 mA for 60 min. WT and knockout bands were detected at 250 and 300 bp, respectively, compared to the standard DNA ladder.

### 2.3. Calculate the Embryonic Age of Mice

Mice were mated the previous day and examined for vaginal plugs the next morning. Mouse embryos with vaginal embolism were counted as E0.

### 2.4. Antibodies

The primary antibodies were rabbit TRIM32 antibody (sc-99011, Santa Cruz, Dallas, TX, USA), mouse TRIM32 antibody (SAB1407164, Sigma, St. Louis, MO, USA), rabbit anti-TBR1 (ab3190, Abcam, Cambridge, UK), rabbit anti- BCL11B (ab28448, Abcam, Cambridge, UK), rabbit anti-CUX1 (sc-13024, Santa Cruz, Dallas, TX, USA), chicken anti-PAX6 (AB_528427, Developmental Studies Hybridoma Bank, Lowa, IA, USA), rabbit anti-TBR2 (ab23345, Abcam, Cambridge, UK), Click-iT™ EdU imaging kit (C10086, Thermo Fisher, Rockford, IL, USA), rabbit anti-PH3 (9713P, Cell Signaling Technology, Boston, MA, USA), rabbit anti-active caspase-3 (C8487, Sigma, St. Louis, MO, USA). The corresponding secondary antibodies conjugated with Alexa fluorophores 488/555/647 (A21202, A31570, A28181, A21206, A31572, A32795, A11039) were from Invitrogen.

### 2.5. Immunofluorescence Staining and Image Analysis

Mice were perfused with ice-cold PBS followed by 4% paraformaldehyde and post-fixed in 4% paraformaldehyde for 4 h followed by dehydration in 10%, 20%, and 30% sucrose, respectively. TRIM32^−/−^ and TRIM32^+/+^ mouse cortex or subventricular zone slices with the same anatomical position were taken for staining. Cryosections were washed three times with PBS containing 0.3% Triton X-100, 10 min each, and then non-specific binding sites were blocked with 10% BSA and goat (or donkey) serum at 10% for 1 h, then overnight incubated with primary antibody at 4 °C. Sections were washed three times in PBS and incubated with appropriate fluorescent secondary antibodies for 2 h at room temperature, then washed three times with PBS and mounted in mounting medium containing DAPI FluoromountG^®^ (010020, Southern Biotech, Birmingham, AL, USA). The stained sections were examined with a confocal laser scanning microscope LSM 700 (Zeiss, Oberkochen, Germany). The same site and area (50 µm × 300 µm) were taken in the stained section and the number of marker cells was calculated. Images were taken with a confocal microscope. For each mouse genotype, we selected anatomical sections from at least three mice for serial analysis. The area of DAPI^+^, TBR1^+^, BCL11B^+^, CUX1^+^, PAX6^+^, TBR2^+^, Edu^+^, PH3^+^, active Caspase3^+^, or Edu^+^/TBR1^+^ cells in each image was quantified using Image J software as described [14].

### 2.6. Edu Pulse Chase

#### 2.6.1. The Ability of NPCs Proliferation

Pregnant mice were injected intraperitoneally with Edu (50 mg/kg body weight) at E14.5 and E16.5 and the offspring were sacrificed after 30 min. Coronal brain sections were taken for Edu staining to analyse cell proliferation.

#### 2.6.2. The Ability of NPCs to Differentiate into TBR1 Positive Neurons

The pregnant mice were injected intraperitoneally with Edu (50 mg/kg body weight) at E13.5, and the offspring were sacrificed at E18.5. Coronal brain sections were stained for Edu and TBR1. The numbers of Edu^+^TBR1^+^ cells were quantified.

### 2.7. Sample Preparation and RNA-seq Analysis

Under anaesthesia with sodium pentobarbital (80 mg/kg) administered by intraperitoneal injection, WT and TRIM32 KO mice were sacrificed at E18.5 and the brain removed. Isolated mice were immediately stored in RNAlater^®^ solution (Ambion, Rockford, IL, USA) at 4 °C. Tissue samples were transferred to −80 °C for storage until analysis after 24 h. RNA was extracted with TRIzol^®^ reagent (Life Technologies, Rockford, IL, USA) and RNeasy kit (Qiagen, Dusseldorf, Germany) for RNA sequencing. The R DESeq2 software package was used to analyse differential expression in RNAseq, and the built-in algorithm in DESeq2 was used for normalization. Pairwise comparisons between the brains of two groups are performed on all genes, and further analysis was performed with a fold change >2 to extract the differentially expressed genes (*p* < 0.05). A total of 39 upregulated genes and 36 down-regulated genes were obtained. KEGG enrichment was performed by the DA-VID webtool (https://david.ncifcrf.gov/home.jsp) on 28 December 2021. The protein-protein interaction network was done by the GeneMANIA webtool (http://genemania.org/) on 14 January 2022.

### 2.8. Statistical Analysis

Values are presented as mean ± SEM. Data were analysed using Student’s *t*-test using SPSS 18.0 software (SPSS, Chicaco, IL, USA). Significance in differences was accepted at *p* < 0.05. * *p* < 0.05; ** *p* < 0.01; *** *p* < 0.001.

## 3. Results

### 3.1. The Expression Pattern of TRIM32 in the Cortex during Embryo

According to previous studies, TRIM32 is expressed by NPCs, including interneuron progenitors within the developing ventricular zone [10,11,14]. In order to determine which types of NPCs express TRIM32, we immunostained for the markers Sox2 (a marker for NPCs), Pax6 (an indicator of radial glial cells) or Tbr2 (an indicator of IPCs) in the dorsal subventricular zone (dSVZ) of E14. 5. It was found that both Sox2^+^ and Pax6^+^ cells expressed TRIM32. In contrast, only a few Tbr2^+^ IPCs expressed TRIM32 (Figure 1A). TRIM32 is primarily expressed in RGCs rather than IPCs in the dSVZ, as shown by these results. In dSVZ cells, TRIM32 was detected in both the cytoplasm and nucleus of RGCs and IPCs. Whereas TRIM32 exhibited nuclear location in the cortical neurons labelled with TBR1, a marker of layer VI neurons, in the cortical plate (CP) at E13.5 and 18.5. However, TRIM32 was transiently expressed in a population of cells below the layer VI cortex at E16.5, where it was located in the cytoplasm (Figure 1B). The dynamic expression of TRIM32 in developing brains suggests it may be an essential regulator of brain development.

### 3.2. TRIM32 Deficiency Mice Exhibit Reduced Size of Brain

At E18.5 (Figure 2A), when neurons are complete, TRIM32^−/−^ mice displayed a smaller brain [16]. To measure the width of the neocortex and cortical plate in the cortex, we used DAPI immunofluorescence staining and Image J to measure the width. According to the results, the width of the neocortex (Figure 2B,C) and the cortical plate (Figure 2D,E) was decreased in the cortex of E18.5 TRIM32^−/−^ mice. These results indicate that TRIM32 regulated the size of developing brain.

### 3.3. TRIM32 Deficiency Results in Reduced Generation of Cortical Neurons in Developing Cortex

Further analysis examined whether deficiency of TRIM32 causes abnormal numbers of pyramidal neurons at different developmental stages using these neuronal markers. TBR1 is expressed in layer VI neurons, BCL11B is expressed in layer V neurons, CUX1 is expressed in layer II-IV neurons [2]. The results showed that layer VI Tbr1^+^ cells were reduced in the cortex of TRIM32^−/−^ mice at E14.5, E16.5 and E18.5 (Figure 3A,B). Layer V BCL11b^+^ cells were downregulated in the cortex of TRIM32^−/−^ mice at E16.5 and E18.5 (Figure 3C,D), while layer II-IV CUX1^+^ neurons in the cortex of TRIM32^−/−^ mice were downregulated at E18.5 and P30 (Figure 3E,F). To further confirm that the decreased number of layer-specific cortical neurons is due to impaired cortical neuron generation, we performed a pulse tracking experiment to label newborn layer V cortical neurons. Pregnant mice were injected intraperitoneally with Edu at E13.5 and mice were scarified at E18.5 and immunostained for Edu and TBR1. Quantitative analysis showed that Edu^+^ had lower numbers of TBR1^+^ in TRIM32^−/−^ VZ/SVZ compared to TRIM32^+/+^ mice (Figure 4A,B). Thus, these results suggest that the absence of TRIM32 leads to reduced formation of pyramidal neurons during brain development. Pyramidal neurons are decreased until adulthood, when brain development is completed, excluding the possibility that TRIM32 deficiency delays the generation of cortical neurons without altering the overall number.

### 3.4. TRIM32 Deficiency Causes Smaller Size of Neural Progenitor Pool

Since TRIM32 deficiency resulted in a decreased number of cortical neurons, we wonder if such an observation is caused by an altered number of progenitor cells. IPC in the dorsal subventricular zone (dSVZ) [17,18]. The latter can also directly generate cortical pyramidal neurons in all layers [19,20,21].

Therefore, we examined whether the number of RGCs and IPCs was altered by the absence of TRIM32 by immunostaining for PAX6 and TBR2, which label RGCs and IPCs, respectively. Results showed that TRIM32^−/−^ mice had decreased numbers of PAX6^+^ RGC (Figure 5A,B) and TBR2^+^ IPC (Figure 5C,D) in VZ/SVZ at E14.5 and E16.5 compared to TRIM32^+/+^ littermates, suggesting that TRIM32 deficiency results in a smaller neural progenitor pool size.

### 3.5. TRIM32 Deficiency Decreases Proliferation and Mitosis of Both RGCs and IPCs

To further investigate the mechanisms underlying the smaller RGC and IPC cluster size caused by the absence of TRIM32, we used Edu to target NPCs in the VZ/SVZ zone at E14.5 and E16.5, respectively to mark. After 30 min, the offspring were sacrificed. Coronal brain sections were taken for Edu staining to analyse NPC proliferation. We found that Edu^+^ cells were reduced in TRIM32^−/−^ VZ/SVZ at E14.5 and E16.5, respectively, compared to TRIM32^+/+^ mice (Figure 6A,B). We also compare the mitotic NPCs in the basal and apical VZ, corresponding to RGC and IPC, respectively. PH3 (mitosis-specific marker) positive cells in both basal VZ and apical VZ showed decreased numbers at E14.5 (Figure 6C,D). In contrast, only PH3^+^ cells in the apical VZ show reduced numbers in E16.5 TRIM32^−/−^ mice compared to those in TRIM32^+/+^ littermates (Figure 6C,E), indicating reduced mitosis of both RGCs and of IPC in developing TRIM32^−/−^ brains. These results indicate that TRIM32 deficiency leads to reduced self-renewal in NPCs.

### 3.6. TRIM32 Deficiency Does Not Affect Apoptosis in Developing Cortex

Apoptosis is an essential mechanism in regulating the number of cortical neurons and NPCs. The brain tends to produce excessive numbers of neurons during development, and some neurons undergo apoptosis at a later stage, ultimately determining the number of neurons in the brain. It has also been reported that the increase in neurons caused by decreased apoptosis causes neuropsychiatric diseases [22,23]. To investigate whether apoptosis is responsible for the reduced numbers of cortical neurons and NPCs, we immunostained Caspase3 to label apoptotic cells in the developing cerebrum. TRIM32^−/−^ mice and their wild-type littermates displayed identical numbers of caspase3^+^ cells in the dorsal telencephalon (Figure 7). Thus, TRIM32 deficiency does not affect the apoptosis of cortical neurons and NPCs in the developing cortex. The decreased number of cortical neurons and NPCs observed in TRIM32/brains was not due to increased apoptosis.

### 3.7. Downstream Signalling Regulated by TRIM32

The mammalian target of rapamycin (mTOR) signalling pathway plays an essential role in brain development [14,24]. mTOR, when overactivated, is one of the central pathways in the etiology of ASD [25,26]. TRIM32 maintained mTOR activity by promoting proteosomal degradation of G protein signalling protein 10 (RGS10). The absence of TRIM32 leads to suppression of mTOR signalling, which in turn leads to increased autophagy, followed by accelerated c-myc degradation [14]. TRIM32 regulates the formation of GABAergic interneurons via this signalling pathway. To further identify additional signalling pathways regulated by the absence of TRIM32, we performed RNAseq analysis on the brains of WT and TRIM32^−/−^ mice. The results showed that Notch signalling was downregulated in TRIM32^−/−^ mice (Figure 8A). Stem cells and neuronal migration in embryonic and adult brain neurogenesis [27]. mTOR is a positive regulator of Notch signalling in human and mouse cells that acts through induction of the STAT3/p63/Jagged signalling cascade [28]. In the downregulated protein interaction diagram, the genes directly related to the Notch pathway are deltx E3 ubiquitin ligase (DTX2) and mastermind-like transcription coactivator 3 (MAML3) (Figure 8B). We were more interested in genes like homeobox A1 (HOXA1) and MAML3 (Figure 8B,C). HOXA1 has been reported to play a crucial role in multiple biological processes and can mediate gene expression and cell differentiation [29,30]. In addition, HOXA1 could be regulated by the AKT/mTOR signalling pathway and the Notch1 signalling pathway [31,32]. MAML3 was part of the Notch1-containing ternary complex in vivo [33], and has been reported to act as a transcriptional coactivator of NOTCH2. Heynen et al. reported that the results indicate an important mechanistic role for MAML3 in retinoic acid-mediated proliferation and differentiation [34]. DTX2 is a homologue of DTX1. The latter regulates Notch signalling in a Su(H)/RBPJ-independent signalling pathway [35]. DTX2 activity is required for neural crest formation, indicating its role in neural development. In addition to Notch signalling, DTX2 also regulates BMP signalling, which plays an essential role in cortical development [35,36]. In conclusion, HOXA1, DTX2 and MAML3 may be involved in mTOR or Notch signalling to affect pyramidal neuron generation. Considering that the interacting proteins can be direct or indirect, this is not clear yet, so we will check these issues in the next investigations.

## 4. Discussion

The imbalance between excitability and inhibitory neural activity (E/I) of brain circuits is a fundamental mechanism for the pathogenesis of ASD. The E/I imbalance could be caused by abnormal generation of GABAergic or pyramidal neurons or their dysfunction. The present study shows that *TRIM32*, an ASD risk gene [12], is an important regulator in the formation of cortical pyramidal neurons. This observation helps explain the E/I imbalance observed in TRIM32^−/−^ mice [37]. MGE-derived GABAergic interneurons begin to produce at day E9.5 and peak at day E13.5 [38]. Pyramidal neurons are mainly produced during the period from E11 to E17 [16]. Interestingly, TRIM32 is expressed in both RGCs located in the dorsal VZ and in interneuron progenitors in L/MGE. Consistent with our previous observation that TRIM32 deficiency causes the loss of multiple subtypes of GABAergic interneurons in the adult brain, the present study observed the loss of pyramidal neurons in the cortex of 1-month-old TRIM32^−/−^ mice as the neuron excitatory pyramidal cells begin their generation and development. Cortical pyramidal neurons are obviously not enough to compensate for the loss of GABAergic interneurons. In addition, the density of neurons can affect the development of neurons, for example, the density of dendritic spines, the arrangement of synapses. In line with this point, indeed, note the dysfunction of TRIM32-deficient synapses, which also exhibited hyperexcitability [37]. Therefore, all these abnormal processes can eventually lead to an E/I imbalance, which in turn causes autism-like behaviours in TRIM32^−/−^ mice [39].

In terms of the cellular mechanism, we have further observed that the absence of TRIM32 affects the self-renewal of both RGCs and IPCs, consistent with the findings that TRIM32 deficiency results in reduced neural proliferation both in vivo and in vitro progenitor cells, including L/MGE parents. Research has shown that some children with ASD have abnormal brain size [40]. Interestingly, in the current study, poor self-renewal in NPCs, including RGC, IPC and L/MGE parents, causes smaller NPC pool size, ultimately resulting in smaller brain size. Using the pulse chase experiment, we also confirmed decreased neuronal differentiation caused by TRIM32 deficiency. Thus, impaired self-renewal coupled with decreased neuronal differentiation contributes to the decreased number of pyramidal neurons in TRIM32-deficient mice.

In terms of the molecular mechanism, our previous results suggest that TRIM32 maintains mTOR activity by promoting RGS10 degradation by proteasomes. Increased mTOR-mediated autophagy promotes c-myc degradation and impairs proliferation of TRIM32-deficient L/MGE precursors [14]. Since TRIM32-deficient RGCs share a similar cellular property to TRIM32-deficient L/MGE progenitors, we propose that these two different progenitors share a similar molecular mechanism. To identify additional downstream signalling pathways caused by TRIM32 deficiency, we performed RNAseq analysis embryonic cortex with TRIM32 deficiency. Notch signalling plays a critical role in neural stem cell maintenance and neurogenesis in both the embryonic and adult brain [27]. Notch is also a substrate for autophagy. Its autophagic degradation is required in stem cell development and neurogenesis [41]. In the down-regulated protein interaction diagram, we identified HOXA1 and MAML3 proteins. HOXA1 acted as a DNA-binding transcription factor that mediated cell differentiation [29,30]. It is worth noting that the Hoxa1 gene is genetically connected to ASD [42,43,44], and Hoxa1 is involved in the proliferation of MIR99 cells by the AKT/mTOR signalling [31]. MAML3 was recognized in vivo as part of the Notch1-containing ternary complex [33], and acted as a transcriptional coactivator for NOTCH2. MAML3 showed an important role in the proliferation and differentiation mechanism [34]. The downregulated protein Rab1A, a conserved small guanosine triphosphatase (GTPase), predominantly regulates the transport of vesicular proteins from the endoplasmic reticulum (ER) to the Golgi apparatus and is involved in mediating Notch signalling, adhesion cell and cell migration [45,46]. Therefore, HOXA1 and MAML3 may be involved in mTOR or Notch signalling to affect pyramidal neuron generation, which will be explored further in the future.

## 5. Conclusions

In summary, this study demonstrates the effects of TRIM32 on the generation of pyramidal neurons in embryonic development and explores their mechanism. In TRIM32^−/−^ mice, NCx and CP were decreased in the cortex, cortex size was decreased, and the number of pyramidal neurons was also decreased. TRIM32 deficiency decreased RGC and IPC cells, proliferating and mitotic neural progenitor cells in VZ/SVZ, decreased cell migration and differentiation in TBR1 positive neuron in cortex, but does not affect TRIM32 apoptosis regulates the generation of cortical pyramidal neurons cells influencing the proliferation, migration and differentiation of RGCs or IPCs and may pass through the mTOR or Notch signalling pathway. The completion of this research provides a theoretical basis for elucidating the mechanism of generation of pyramidal neurons in the embryonic period and provides strong data support for exploring the onset and development of related neurodevelopmental diseases.

## Figures and Tables

**Figure 1 cells-11-00449-f001:**
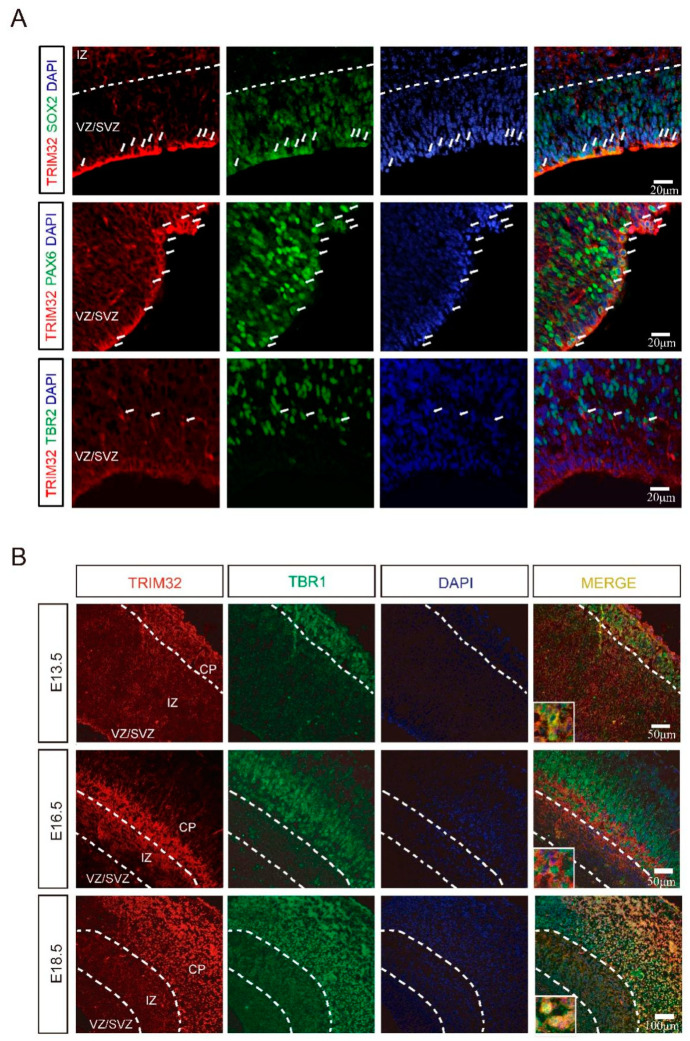
The expression pattern of TRIM32 in the developing cortex. (**A**) The coronal sections of telencephalon from E14.5 mice were immunostained for TRIM32, SOX2/PAX6/TBR2 and DAPI. Scale bars = 20 μm. (**B**) The coronal sections of telencephalon from E13.5, E16.5 and E18.5 mice were immunostained for TRIM32, TBR1 and DAPI. Scale bars = 50 μm (upper and middle panel); Scale bar = 100 μm (bottom panel).

**Figure 2 cells-11-00449-f002:**
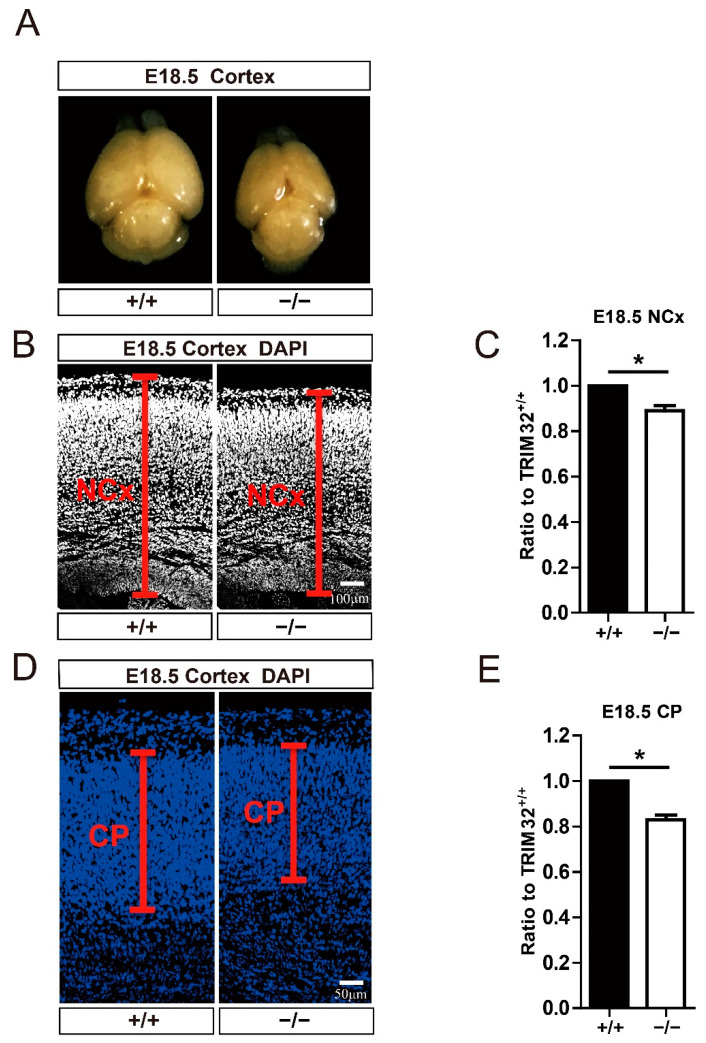
Brain size and cortical width of E18.5 TRIM32^−/−^ mice and TRIM32^+/+^ littermates. (**A**) Representative images of brain size. (**B**,**D**) The dorsal telencephalon was stained for DAPI. The width of neocortex (NCx) and cortical plate (CP) was indicated. Scale bars = 100 μm. (**C**,**E**) The relative width of TRIM32^−/−^ neocortex and cortical plate. The width of TRIM32^+/+^ was normalized to 1.0. *n* = 15 slices for 3 mice/genotype. Scale bars = 50 μm. * *p* < 0.05.

**Figure 3 cells-11-00449-f003:**
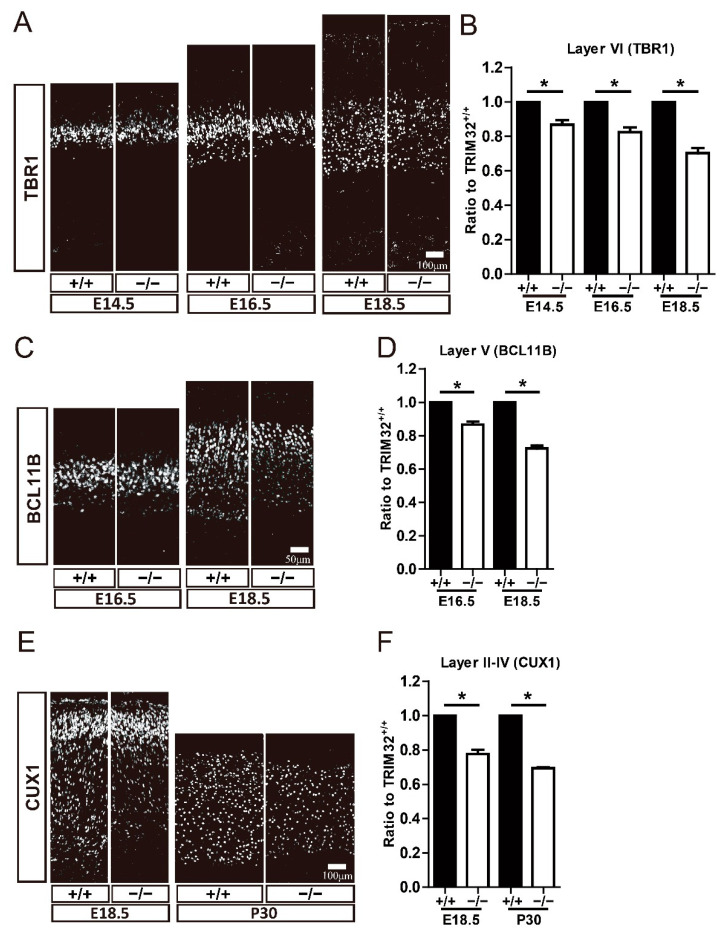
TRIM32^−/−^ mice exhibited decreased numbers of both deep-and upper-layer cortical neurons. The coronal sections of cerebral cortex at distinct developmental stages as indicated were immunostained for TBR1 (**A**), BCL11B (**C**) and CUX1 (**E**). Relative density of TBR1^+^ (**B**), BCL11b^+^ (**D**) and CUX1^+^ (**F**) cells in TRIM32^−/−^ brains. The density of above cells in TRIM32^+/+^ brains were normalized to 1.0. *n* = 15 slices for 3 mice/genotype. Scale bars = 100 μm (**A**,**E**); Scale bars = 50 μm (**C**). * *p* < 0.05.

**Figure 4 cells-11-00449-f004:**
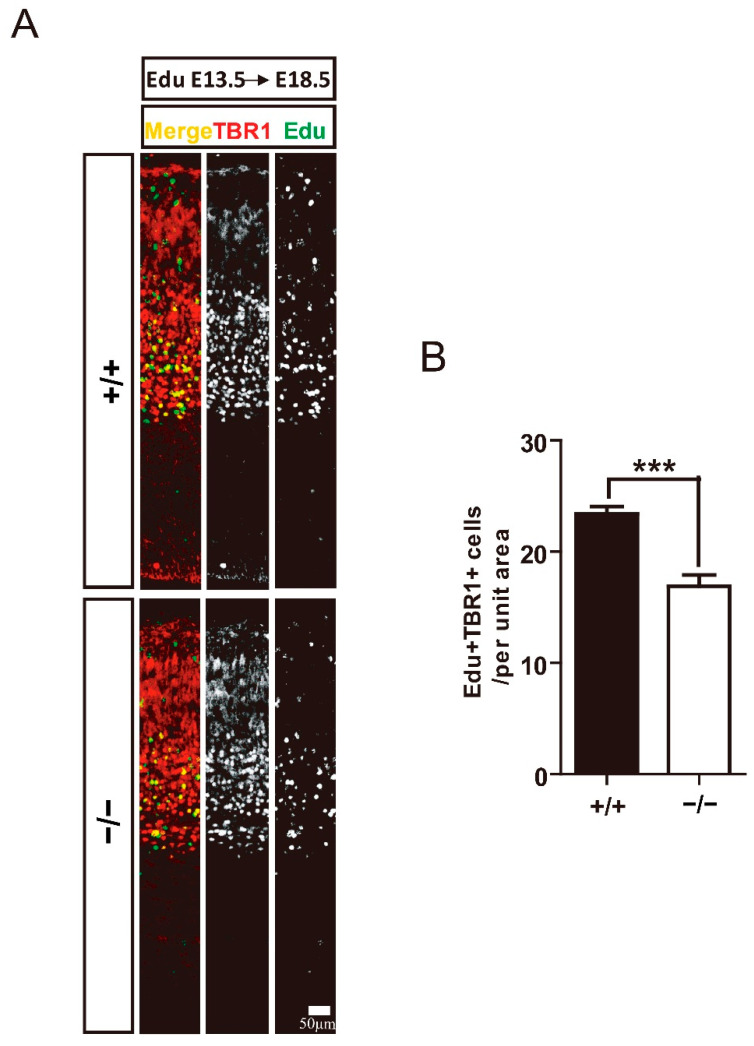
Impaired generation of layer VI cortical neurons in TRIM32^−/−^ mice. Edu were injected intraperitoneally at E13.5. The mice were scarified at E18.5. The coronal sections were immunostained for Edu and TBR1 (**A**). Density of EdU^+^TBR1^+^ cells (**B**). *n* = 15 slices for 3 mice/genotype. Scale bars= 50 μm. *** *p* < 0.001.

**Figure 5 cells-11-00449-f005:**
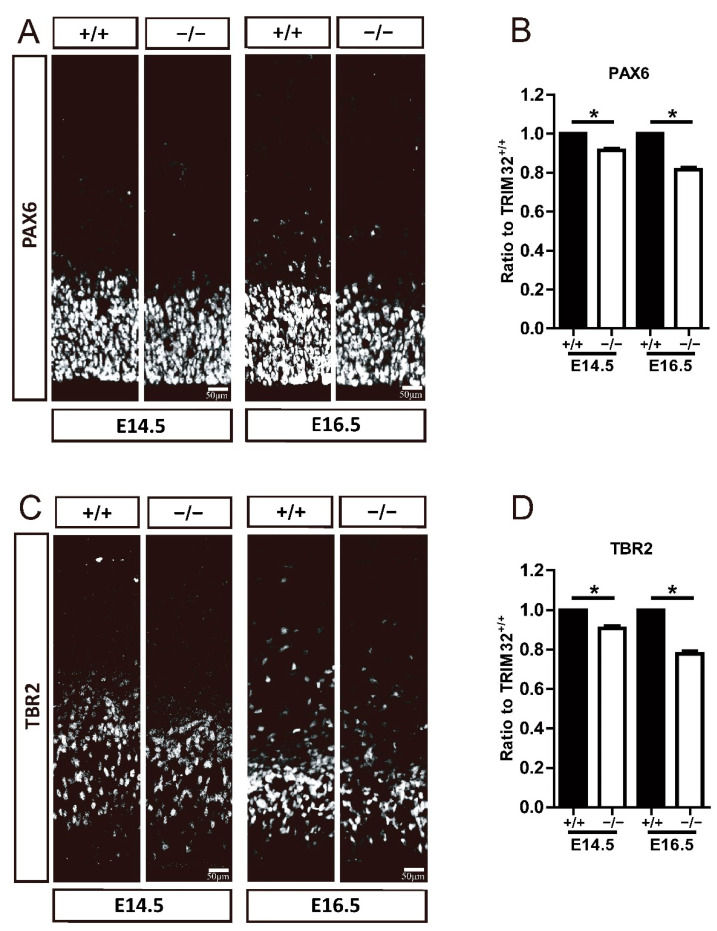
TRIM32^−/−^ mice exhibited decreased numbers of RGCs and IPCs. The coronal sections of cerebral cortex at distinct E14.5 and E16.5 were immunostained for either PAX6 (**A**) or TBR2 (**C**). Relative density of PAX6^+^ (**B**) and TBR2^+^ cells (**D**) in TRIM32^−/−^ brains. The density of above cells in TRIM32^+/+^ brains were normalized to 1.0. *n* = 15 slices for 3 mice/genotype. Scale bars = 50 μm. * *p* < 0.05.

**Figure 6 cells-11-00449-f006:**
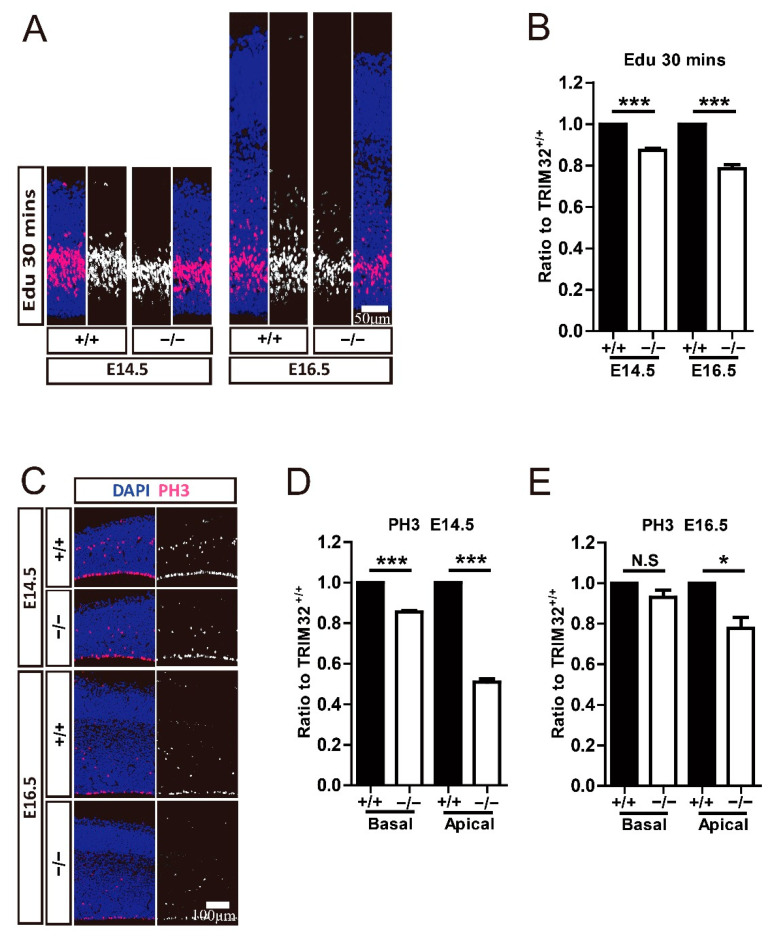
TRIM32^−/−^ mice exhibited decreased proliferation and mitosis. The mice were scarified 30 min after being injected intraperitoneally with Edu at either E14.5 or E16.5. The coronal cortical sections were immunostained for either Edu (**A**) or PH3 (**C**) and DAPI. Relative density of Edu^+^ (**B**) or PH3^+^ cells (**D**,**E**) in TRIM32^−/−^ brains. The density of above cells in TRIM32^+/+^ brains were normalized to 1.0. *n* = 15 slices for 3 mice/genotype. Scale bars = 50 μm (**A**); Scale bars = 100 μm (**C**). * *p* < 0.05; *** *p* < 0.001; N.S: no significance.

**Figure 7 cells-11-00449-f007:**
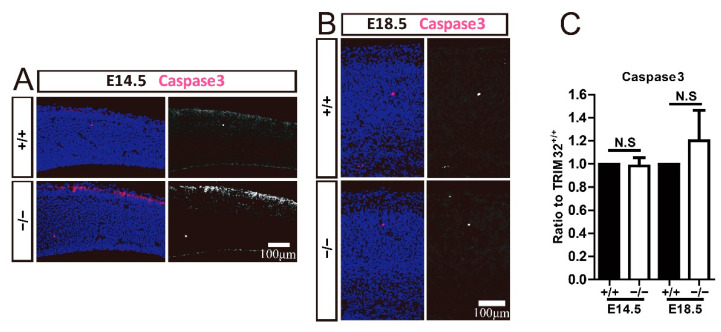
TRIM32 deficiency does not affect apoptosis. The coronal cortical sections were immunostained for Caspase3 and DAPI at E14.5 and E18.5 (**A**,**B**). Relative density of Caspase3^+^ cells (**C**) in TRIM32^−/−^ brains. The density of above cells in TRIM32^+/+^ brains were normalized to 1.0. *n* = 15 slices for 3 mice/genotype. Scale bars = 100 μm. N.S: no significance.

**Figure 8 cells-11-00449-f008:**
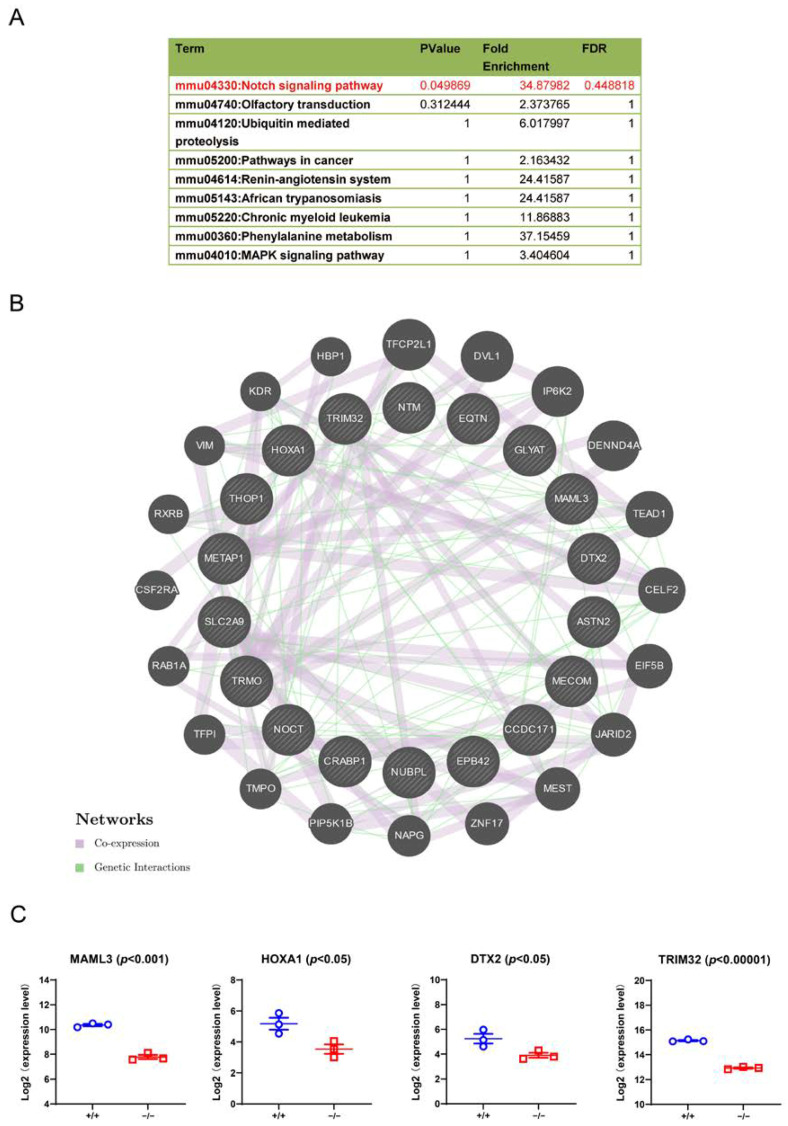
TRIM32 deficiency impairs Notch signalling pathway. (**A**) The enriched pathways for downregulated genes in TRIM32^−/−^ mice. (**B**) The downregulated protein interaction diagram. The inner circle represents all identifiable down-regulated genes, and the outer circle represents the genes that might be predicted to interact with these down-regulated genes. The types of interactions between genes are illustrated. (**C**) Expression levels of MAML3, HOXA1, DTX2, and TRIM32 were compared in TRIM32^+/+^ mice (+/+) andTRIM32^−/−^ mice (−/−).

## Data Availability

All data generated or analysed during this study are available from the corresponding author upon reasonable request.

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
