# Peer review of "TRIM32 Deficiency Impairs the Generation of Pyramidal Neurons in Developing Cerebral Cortex"

_cells, 2022, doi:10.3390/cells11030449_

Round 1

Reviewer 1 Report

In this manuscript, Sun and colleagues investigated the roles of TRIM32 in neocortical development by analyzing TRIM32-deficent mice. They found that TRIM32 regulates the proliferation of NPCs via Notch signaling, thus modulates the neurogenesis and cortical development of mouse brain. The study is well designed and the manuscript is well written. There are some concerns authors should address to improve the current manuscript.

  1. Fig 1, authors should also present the costaining of TRIM32 and NPC marker Nestin to describe the expression of TRIM32 in RGCs.

  1. Line 62, “RGCs and OPCs” should be “RGCs and IPCs”

Author Response

Response to Reviewer 1 Comments

In this manuscript, Sun and colleagues investigated the roles of TRIM32 in neocortical development by analyzing TRIM32-deficent mice. They found that TRIM32 regulates the proliferation of NPCs via Notch signaling, thus modulates the neurogenesis and cortical development of mouse brain. The study is well designed and the manuscript is well written. There are some concerns authors should address to improve the current manuscript.

Point 1: Fig 1, authors should also present the costaining of TRIM32 and NPC marker Nestin to describe the expression of TRIM32 in RGCs.

Response 1: We thank the reviewer’s suggestion. We have showed this result in Figure 1.  

Point 2: Line 62, “RGCs and OPCs” should be “RGCs and IPCs”

Response 2: We thank the reviewer to point out this question. We have modified it in page 2, line 63.

Reviewer 2 Report

The study provided by Yan-Yun Sun and coauthors is devoted to TRIM32 deficit effect on the generation of cortical pyramidal neurons in mice. The manuscript provides important information about the role of TRIM32 in neuronal network development and maturation. Experimental design is appropriate; however, there is a major concern about the statistical analysis of the obtained results. As the authors described in the figure legends, they used only 3 mice in each experimental series and performed Student`s test to compare the differences. How did the authors evaluate normality using only 3 repeats? So small sample size requires non-parametric tests. The significance of some differences is very low (*), and non-parametric tests may give another result. If so, some conclusions will become ambiguous.

Minor points:

  • Please, clearly describe what indicates each antibody's positive staining (for example, PH3)? Marker for what?
  • Change the size and colors of some titles/abbreviations in figures (Figure 2B, Figure 4A) because the words are almost invisible. Please, enlarge the scale bars in panels and reorganize or combine some figures. Figures 4 and 8 seem disproportional, it is better to revise them.

Author Response

Response to Reviewer 2 Comments

Point 1: The study provided by Yan-Yun Sun and coauthors is devoted to TRIM32 deficit effect on the generation of cortical pyramidal neurons in mice. The manuscript provides important information about the role of TRIM32 in neuronal network development and maturation. Experimental design is appropriate; however, there is a major concern about the statistical analysis of the obtained results. As the authors described in the figure legends, they used only 3 mice in each experimental series and performed Student`s test to compare the differences. How did the authors evaluate normality using only 3 repeats? So small sample size requires non-parametric tests. The significance of some differences is very low (*), and non-parametric tests may give another result. If so, some conclusions will become ambiguous.

Response 1: We thank the reviewer for pointing out the concern. In the current study, although we used 3 mice in each experimental series, we collected at least 5 brain slices from each mouse for completing our statistics. Thus, n=15 slices for 3 mice/genotype in our data, which I have modified in Figure legends.

Point 2: Please, clearly describe what indicates each antibody's positive staining (for example, PH3)? Marker for what?

Response 2: We thank the reviewer for this suggestion. In the revised manuscript, we have explained the immmunostainning in more details. Please see the line 258-259 in page 9.

Point 3: Change the size and colors of some titles/abbreviations in figures (Figure 2B, Figure 4A) because the words are almost invisible. Please, enlarge the scale bars in panels and reorganize or combine some figures. Figures 4 and 8 seem disproportional, it is better to revise them.

Response 3: We thank the reviewer for this suggestion. The figures are revised.

Reviewer 3 Report

Overall, this is an interesting research article in which Sun et al., study the role of TRIM32 on the embryonic development of pyramidal neurons. The authors conclude that TRIM32 deficiency impairs cortical pyramidal neurons generation driving to a smaller brain size. They propose that the diminution in the number of pyramidal neurons is due to a decrease in the proliferation, migration and differentiation of RGCs or IPCs through mTOR or Notch pathway, rather than to an increase on pyramidal neurons apoptosis.

However, I have several comments:

  1. Why did you use different embryonic stage in the different experiments? I mean, to study the expression of TRIM32 in the cortex you use E13.5, E16.5 and E18.5; Figure 1, while in other experiments such us the experiments in which you study the generation of cortical neurons you use E14.5, E16.5, E18,5; Figure 3.
  2. In the same way, did you check for all the neuronal markers the embryonic stage E30? Also, did you only show in the figures (at least in figure 3) the results that were statistically significant? It is not clear for my if you check in all the experiments the embryonic stage E13.4 or 14.5, E16.5, E 18.5 and E30. Please clarify on the text.
  3. Authors conclude that TRIM32 deficiency does not affect apoptosis. However, in Figure 7 at E18,5 stage is seem that there is a higher density of caspase 3 cells in TRIM32-/- and although is not statistical (maybe in part to the high deviation) the sample is too small to totally discard the influence of apoptosis on the pyramidal pool size. You should or increase the sample or at least discuss more this issue.
  4. Authors state that the diminution of excitatory pyramidal neurons cause by TRIM32 deficiency could led to the excitatory/inhibitory imbalance underlying neurodevelopmental disorders such autism disorders. This imbalance drive in these disorders to an hyperexcitability, but I don’t understand the mechanism by which the diminution of excitatory pyramidal neurons could led to this situation. You should explain this issue with more detail in discussion. In general, discussion could be improved. 

Other minor comments:

5.You must indicate in Material and Methods the concentration of sodium pentobarbital that you have use to anesthetize the animals.

6.What does mean the abbreviature OPCs in line 62 and 82? I think is IPCs.

7.Please, check the sentence that goes from line 91 to 93, I think you must remove "all litter control mice" or change t

8. Change” form to from” in line 119.

9. Delete one “were” in line 143.

Author Response

Response to Reviewer 3 Comments

Overall, this is an interesting research article in which Sun et al., study the role of TRIM32 on the embryonic development of pyramidal neurons. The authors conclude that TRIM32 deficiency impairs cortical pyramidal neurons generation driving to a smaller brain size. They propose that the diminution in the number of pyramidal neurons is due to a decrease in the proliferation, migration and differentiation of RGCs or IPCs through mTOR or Notch pathway, rather than to an increase on pyramidal neurons apoptosis. However, I have several comments:

Point 1: Why did you use different embryonic stage in the different experiments? I mean, to study the expression of TRIM32 in the cortex you use E13.5, E16.5 and E18.5; Figure 1, while in other experiments such us the experiments in which you study the generation of cortical neurons you use E14.5, E16.5, E18,5; Figure 3.

Response 1: We understand the reviewer’s concern. In the developing brain, distinct cortical neurons are produced in a temporal order. For example, TBR1+ layer VI neurons and BCL11B+ layer V neurons complete their predominate generation by E14.5 and E16.5, respectively. While CUX1+ layer II-IV neurons are produced at E18.5. Thus, we chose at E14.5, E16.5 and E18.5, to observe the numbers of the aboved cortical neurons. We agree with the reviewer that it is better to analyze the expression of TRIM32 at the same developmental stage. However, due to the limited available brain slices, we have used E13.5, E16.5 and E18.5 in figure 1. The generation of cortical neurons starts as earlier as E11.5. By E14, neurogenesis reaches its peaks. We propose to investigate the expression of TRIM32 in cortical neurons in figure 1. Thus, in this context, E13.5 and E14.5 makes subtle differences. To further observe the expression of TRIM32 in neural progenitor cells, we have stained for TRIM32 with Sox2 and Pax6, two markers for neural progenitor cells, at E14.5. Please see figure 1A in the revised manuscript.

Point 2: In the same way, did you check for all the neuronal markers the embryonic stage E30? Also, did you only show in the figures (at least in figure 3) the results that were statistically significant? It is not clear for my if you check in all the experiments the embryonic stage E13.4 or 14.5, E16.5, E 18.5 and E30. Please clarify on the text.

Response 2: TBR1+ layer VI neurons and BCL11B+ layer V neurons complete their generation by E18.5. Thus, in terms of these two deep-layer cortical neurons, we have examined their numbers at E18.5, rather than later on stage. However, the generation of CUX1+layer II-IV neurons is not completed at18.5. Thus, we investigated numbers of CUX1+ neuron at P30, when CUX1+neurons finished their production completely.

Point 3: Authors conclude that TRIM32 deficiency does not affect apoptosis. However, in Figure 7 at E18,5 stage is seem that there is a higher density of caspase 3 cells in TRIM32-/- and although is not statistical (maybe in part to the high deviation) the sample is too small to totally discard the influence of apoptosis on the pyramidal pool size. You should or increase the sample or at least discuss more this issue.

Response 3: We understand the reviewer’s concern. We used 3 mice in each experimental series, while each mouse having at least 5 brain slices for completing our statistics. So in our data, n=15 slices for 3 mice/genotype, which I have modified in Figure legends. In addition, our previous study also observed TRIM32 deficiency does not affect apoptosis in both in vivo and in vitro [1].

Point 4: Authors state that the diminution of excitatory pyramidal neurons cause by TRIM32 deficiency could led to the excitatory/inhibitory imbalance underlying neurodevelopmental disorders such autism disorders. This imbalance drive in these disorders to an hyperexcitability, but I don’t understand the mechanism by which the diminution of excitatory pyramidal neurons could led to this situation. You should explain this issue with more detail in discussion. In general, discussion could be improved.

Response 4: We appreciate the reviewer’s suggestion. Our previous study observes impaired generation of inhibitory GABAergic interneurons in TRIM32-deficiency mice. The present study observes a decreased production of pyramidal cortical neurons. The reduced numbers of pyramidal cortical neurons are obviously not enough to complement the loss of GABAergic interneurons. Moreover, the density of neurons may affect the development of neurons, for example, the density of dendritic spines, the assembly of synapses. In agree with this point, we indeed observe dysfunction of TRIM32-deficient synapses, which also exhibited hyperexcitability [2]. Thus, all these abnormal processes may eventually lead to the excitatory/inhibitory imbalance caused by TRIM32 deficiency. We have revised the discussion in the revised manuscript. Please see Line 338-344 in page 12-13.

Point 5: You must indicate in Material and Methods the concentration of sodium pentobarbital that you have use to anesthetize the animals.

Response 5: We thank the reviewer for pointing out it. We have modified it in page 4, line 150.

Point 6: What does mean the abbreviature OPCs in line 62 and 82? I think is IPCs.

Response 6: We are sorry for this mistake. We have modified it in page 2, line 63 and 83.

Point 7: Please, check the sentence that goes from line 91 to 93, I think you must remove "all litter control mice" or change it.

Response 7: We are sorry for this mistake. We have modified it in page 2, line 93-94.

Point 8: Change” form to from” in line 119.

Response 8: We are sorry for this mistake. We have modified it in page 3, line 120.

Point 9: Delete one “were” in line 143.

Response 9: We are sorry for this mistake. We have modified it in page 3, line 144.

References:

  1. Zhu, J. W.; Zou, M. M.; Li, Y. F.; Chen, W. J.; Liu, J. C.; Chen, H.; Fang, L. P.; Zhang, Y.; Wang, Z. T.; Chen, J. B., et al. Absence of TRIM32 Leads to Reduced GABAergic Interneuron Generation and Autism-like Behaviors in Mice via Suppressing mTOR Signaling. Cereb Cortex 2020, 30, 3240-58, doi:10.1093/cercor/bhz306.
  2. Ntim, M.; Li, Q. F.; Zhang, Y.; Liu, X. D.; Li, N.; Sun, H. L.; Zhang, X.; Khan, B.; Wang, B.; Wu, Q., et al. TRIM32 Deficiency Impairs Synaptic Plasticity by Excitatory-Inhibitory Imbalance via Notch Pathway. Cerebral Cortex 2020, 30, 4617-32, doi:10.1093/cercor/bhaa064.

Reviewer 4 Report

I have reviewed the manuscript entitled: TRIM32 Deficiency Impairs the Generation of Pyramidal Neurons in Developing Cerebral Cortex, where the authors characterize the expression patterns and phenotypic effects of a TRIM32 KO mouse.

The authors replicates in vitro and in vivo observations by Schwamborn et al. 2009 that TRIM32 inhibits proliferation and is required for neuronal differentiation. They also provide bulk RNA-seq comparison and highlight alterations in NOTCH signalling.

I find that Figure 1 could be improved to be more convincing. It is not clear which kind of cells show expression of TRIM32 in the SVZ at E16.5. Co-staining with a marker of neural progenitors, apical, basal and IP, could be informative. A maximization of the confocal could also help identifying the cytoplasmatic location. Same with TBR1 staining, which is claimed to be in the nucleus, that could be better appreciated zooming in particular cells. Also, please indicate layers in the plot.

TRIM32 has been previously shown to play a role in generation of interneurons. Counting interneurons in the cortex would be confirmatory of that observation.

Expression of TRIM32 in the developing mouse brain was previously studied in Schwamborn et al. 2009. There seems to be some apparent discrepancies with the present study that should be addressed. First, at E12.5 Trim32 is notably expressed in the cytoplasm of progenitor cells, but in the present study the authors show background staining at E13.5 in VZ and CP. Second, at E14.5 Schwamborn et al. showed that Trim32 is already expressed in the cortical plate, and in the cytoplasm, not nuclei; but in here we do not see Trim32 in the CP at E16.5. These at least three discrepancies need to be directly tackled.    

Finally, the presentation of RNA-seq results is not convincing. The authors should explain how many genes are found differentially expressed and provide tables with all DEX genes and all enriched pathways. Only NOTCH is shown, but how many genes in the NOTCH pathways are driving this enrichment? Figure 8A is unnecessary, as it only shows one pathway. Does count = 2 means that only 2 Notch genes were found downregulated? Also, discuss to which degree DEX genes could correspond to variation in cell type composition, rather than true expression differences, given the fact that cell composition is altered in the KO, as shown throughout the paper.

Please provide details for how the “downregulated protein interaction diagram” in figure 8b has been performed and make sure it is reproducible.

Author Response

Response to Reviewer 4 Comments

I have reviewed the manuscript entitled: TRIM32 Deficiency Impairs the Generation of Pyramidal Neurons in Developing Cerebral Cortex, where the authors characterize the expression patterns and phenotypic effects of a TRIM32 KO mouse .The authors replicates in vitro and in vivo observations by Schwamborn et al. 2009 that TRIM32 inhibits proliferation and is required for neuronal differentiation. They also provide bulk RNA-seq comparison and highlight alterations in NOTCH signalling.

Point 1: I find that Figure 1 could be improved to be more convincing. It is not clear which kind of cells show expression of TRIM32 in the SVZ at E16.5. Co-staining with a marker of neural progenitors, apical, basal and IP, could be informative. A maximization of the confocal could also help identifying the cytoplasmatic location. Same with TBR1 staining, which is claimed to be in the nucleus, that could be better appreciated zooming in particular cells. Also, please indicate layers in the plot.

Response 1: We appreciate the reviewer’s suggestion. We have immunostained for TRIM32 and sox2, pax6 (two markers for neural progenitor cells), Tbr2 (a marker for intermediate progenitor cells) following the reviewer’s suggestion. Please see figure 1A in the revised manuscript.

Point 2: TRIM32 has been previously shown to play a role in generation of interneurons. Counting interneurons in the cortex would be confirmatory of that observation.

Response 2: we appreciated with the reviewer’s suggestion. The numbers of subtypes of GABAergic interneurons have been examined in our previous study [1].

Point 3: Expression of TRIM32 in the developing mouse brain was previously studied in Schwamborn et al. 2009. There seems to be some apparent discrepancies with the present study that should be addressed. First, at E12.5 Trim32 is notably expressed in the cytoplasm of progenitor cells, but in the present study the authors show background staining at E13.5 in VZ and CP. Second, at E14.5 Schwamborn et al. showed that Trim32 is already expressed in the cortical plate, and in the cytoplasm, not nuclei; but in here we do not see Trim32 in the CP at E16.5. These at least three discrepancies need to be directly tackled.

Response 3: We have observed that TRIM32 is expressed in the cytoplasm of neural progenitor cells in the present study and our previous study [1], while TRIM32 is located in the nuclei of differentiated neurons. These results is consistent with Hilljie et al., 2011[2] published by Schwamborn’s group as well. Consistent with Schwamborn et al., 2009, we also observed TRIM32 expression in the CP at E13.5 and E18.5. However, we observed that TRIM32 exhibited a transient expression in a population of cells below layer VI cortex, where it located in the cytoplasm (Figure 1B). These observations indicate a dynamic expression of TRIM32 in developing brains, further emphasize its potential role in brain development. We describe the results in more details in the revised manuscript. In addition, we incorporated the location of TRIM32 in neural progenitor cells in the revised manuscript, where cytoplasmic TRIM32 was clearly shown (Figure 1A).

Point 4: Finally, the presentation of RNA-seq results is not convincing. The authors should explain how many genes are found differentially expressed and provide tables with all DEX genes and all enriched pathways. Only NOTCH is shown, but how many genes in the NOTCH pathways are driving this enrichment? Figure 8A is unnecessary, as it only shows one pathway. Does count = 2 means that only 2 Notch genes were found downregulated? Also, discuss to which degree DEX genes could correspond to variation in cell type composition, rather than true expression differences, given the fact that cell composition is altered in the KO, as shown throughout the paper.

Response 4: we appreciate the reviewer’s suggestion. We revised figure 8 where Notch signaling pathway and the expression of their individual related genes are shown. We also discussed the contribution of DEX genes to the reduced pyramidal neurons caused by TRIM32 deficiency. Please see line 290-324 in page 11-12.

Point 5: Please provide details for how the “downregulated protein interaction diagram” in figure 8b has been performed and make sure it is reproducible.

Response 5: We have updated this in Figure 8B.

References:

  1. Zhu, J. W.; Zou, M. M.; Li, Y. F.; Chen, W. J.; Liu, J. C.; Chen, H.; Fang, L. P.; Zhang, Y.; Wang, Z. T.; Chen, J. B., et al. Absence of TRIM32 Leads to Reduced GABAergic Interneuron Generation and Autism-like Behaviors in Mice via Suppressing mTOR Signaling. Cereb Cortex 2020, 30, 3240-58, doi:10.1093/cercor/bhz306.
  2. Hillje, A. L.; Worlitzer, M. M. A.; Palm, T. and Schwamborn, J. C. Neural Stem Cells Maintain Their Stemness through Protein Kinase C zeta-Mediated Inhibition of TRIM32. Stem Cells 2011, 29, 1437-47, doi:10.1002/stem.687.

Reviewer 5 Report

The article: TRIM32 Deficiency Impairs the Generation of Pyramidal Neurons in Developing Cerebral Cortex

The article demonstrates the role of TRIM32 in the cortical neuronal generation.

Major comments:

The RNA-seq result showed that there was a downregulation of the notch pathway.

This result needs to be validated by PCR? Or even at the protein level by western. This also applies to the other interacting genes: HOXA and RAB1A.

no validation is performed, it should be on gene and protein levels

Better images of the immune stains can be added. The resolution is low,

For the immunostaining, since they are looking for neuronal degeneration, doublecortin can be stained to quantify neural precursor cells and immature neurons, for clear validation of neuronal generation.

Minor comments:

Be consistent in the use of acronyms: For instance: for NPCs: sometimes they use neural progenitor cells, and others neural precursor cells.

Most of the sentences need to be modified. The sentence structure and choice of words.

Punctuations need to be reviewed, as well as tenses used in the manuscript.

The authors suggested that “the impaired generation of excitatory pyramidal neurons may explain hyperexcitability observed in TRIM32-deficient mice” Can they expand on this further? because hyperexcitability is usually observed when there is a loss of inhibitory neurons and not excitatory neurons.

Author Response

Response to Reviewer 5 Comments

The article: TRIM32 Deficiency Impairs the Generation of Pyramidal Neurons in Developing Cerebral Cortex.The article demonstrates the role of TRIM32 in the cortical neuronal generation. Major comments:

Point 1: The RNA-seq result showed that there was a downregulation of the notch pathway.

This result needs to be validated by PCR? Or even at the protein level by western. This also applies to the other interacting genes: HOXA and RAB1A.

no validation is performed, it should be on gene and protein levels

Response 1: We appreciate the reviewer’s suggestion. We will verify the levels of these genes with qPCR and WB in future. To clearly shown the change of these genes in absence of TRIM32, we have shown their relative expression levels in the revised manuscript (Figure 8C).

Point 2: Better images of the immune stains can be added. The resolution is low

Response 1: We have updated the figures.

Point 3: For the immunostaining, since they are looking for neuronal degeneration, doublecortin can be stained to quantify neural precursor cells and immature neurons, for clear validation of neuronal generation.

Response 3: We understand the reviewer’s concern. We have labeled RGCs, IPCs and distinct layers of cortical neurons by immunostaining their transcriptional markers. One of the advantages of these transcriptional markers is that they are convenient for counting the cell numbers, despite of their failure in showing the cellular morphology. We have confirmed that the decreased numbers of NPCs and cortical neurons are unlikely due to apoptosis. In the earlier developmental stage, quite few neurodegenerations occur in the brains. We appreciate the reviewer’s suggestion that using doublecortin staining to confirm this point. However, we failed to recollect embryonic brains from TRIM32-deficient and their littermate wild type mice in the limited revision time. We will perform this experiment in future when we investigate how TRIM32 affects neuronal development.

Minor comments:

Point 4: Be consistent in the use of acronyms: For instance: for NPCs: sometimes they use neural progenitor cells, and others neural precursor cells.

Response 4: We apologize for the mistakes. We have corrected this problem in the revised manuscript.

Point 5: Most of the sentences need to be modified. The sentence structure and choice of words. Punctuations need to be reviewed, as well as tenses used in the manuscript.

Response 5: We thank the reviewer for pointing out it.  We have invited a native English speaker to go through the manuscript and polish the language.

Point 6: The authors suggested that “the impaired generation of excitatory pyramidal neurons may explain hyperexcitability observed in TRIM32-deficient mice” Can they expand on this further? because hyperexcitability is usually observed when there is a loss of inhibitory neurons and not excitatory neurons.

Response 6: We appreciate the reviewer’s suggestion. Our previous study observes impaired generation of inhibitory GABAergic interneurons in TRIM32-deficiency mice. The present study observes a decreased production of pyramidal cortical neurons. The reduced numbers of pyramidal cortical neurons are obviously not enough to complement the loss of GABAergic interneurons. Moreover, the density of neurons may affect the development of neurons, for example, the density of dendritic spines, the assembly of synapses. In agree with this point, we indeed observe dysfunction of TRIM32-deficient synapses, which also exhibited hyperexcitability [1]. Thus, all these abnormal processes may eventually lead to the excitatory/inhibitory imbalance caused by TRIM32 deficiency. We have revised the discuss in the revised manuscript. Please see line 338-344 in page 12-13.

References:

  1. Ntim, M.; Li, Q. F.; Zhang, Y.; Liu, X. D.; Li, N.; Sun, H. L.; Zhang, X.; Khan, B.; Wang, B.; Wu, Q., et al. TRIM32 Deficiency Impairs Synaptic Plasticity by Excitatory-Inhibitory Imbalance via Notch Pathway. Cerebral Cortex 2020, 30, 4617-32, doi:10.1093/cercor/bhaa064.

Round 2

Reviewer 1 Report

Authors have addressed all my concerns.

Author Response

We thank the reviewer for the nice comment. Your suggestions and comments helped a lot to improve our manuscript.

Reviewer 2 Report

Most of my comments have been addressed.

Author Response

We appreciate the reviewer’s suggestions and comments, that helped a lot to improve our manuscript.

Reviewer 3 Report

From my point of view, the manuscript has been sufficiently improved to warrant publication in Cells.

Author Response

(The authors gave the same response as above.)

Reviewer 4 Report

I have reviewed authors response and changes. I appreciate the addition of markers for NPC in figure 1A. However, I failed to see much colocalization of SOX2 and TRIM32, maybe a magnification or arrows should help, as in the case of TBR2. As I suggested in my previous revision, in figure 1, please add the layers (VZ, SVZ, IZ, CP).  Also please show the pictures of the entire cortical wall from ventricle to pia in all panels, and a maximization of the colocalizations if needed. That would really help visualizing the transitions, what is transient and what is constant in the same structure through developmental time.

The section of RNA-seq still needs to be improved in my opinion. From the extraction of RNA to the DESeq2 analyses there is a gap in methods on how the authors sequenced the samples. Please provide information on how many genes were up or down-regulated in the KO vs WT at which FDR. Also, provide details in methods on how you produced the interaction diagram, what kind of interaction you are displaying, how many genes you introduced, etc. The RNA-seq analysis is hardly reproducible with the data provided  The manuscripts states that: “All data generated or analyzed during this study are included in thispublished (sic) article.” But I do not see the link to the GEO with the RNA-seq fastq, count matrices and complete set of DEX genes.

Line 147. Scarified at E18.5, rather than P18.5, perhaps?

Author Response

Point 1: I have reviewed authors response and changes. I appreciate the addition of markers for NPC in figure 1A. However, I failed to see much colocalization of SOX2 and TRIM32, maybe a magnification or arrows should help, as in the case of TBR2. As I suggested in my previous revision, in figure 1, please add the layers (VZ, SVZ, IZ, CP). Also please show the pictures of the entire cortical wall from ventricle to pia in all panels, and a maximization of the colocalizations if needed. That would really help visualizing the transitions, what is transient and what is constant in the same structure through developmental time.

Response 1: We understand the reviewer’s concern. We have used arrows to mark the colocalization of SOX2/PAX6/TBR2 and TRIM32 in Figure 1A and added a maximization of the colocalizations in Figure 1B. We also have added the layers (VZ, SVZ, IZ, CP) in Figure 1. Please see the updated figure 1.

Point 2: The section of RNA-seq still needs to be improved in my opinion. From the extraction of RNA to the DESeq2 analyses there is a gap in methods on how the authors sequenced the samples. Please provide information on how many genes were up or down-regulated in the KO vs WT at which FDR. Also, provide details in methods on how you produced the interaction diagram, what kind of interaction you are displaying, how many genes you introduced, etc. The RNA-seq analysis is hardly reproducible with the data provided  The manuscripts states that: “All data generated or analyzed during this study are included in thispublished (sic) article.” But I do not see the link to the GEO with the RNA-seq fastq, count matrices and complete set of DEX genes.

Response 2: We are sorry for not make it clear. We have modified the methods in 2.7 Sample preparation and RNA-seq analysis in page 4, line 154-160, the figure 8B and the figure 8 legend in page 12, line 314-319. And all our data generated or analyzed during this study are available from the corresponding author upon reasonable request. We have modified it in page 14, line 405-406.

Point 3: Line 147. Scarified at E18.5, rather than P18.5, perhaps?

Response 3: We are sorry for this mistake. We have modified it in page 4, line 145.

Reviewer 5 Report

There should be validation of the gene expression 

the notch pathway and other genes

Author Response

We understand the reviewer’s concern and appreciate the reviewer’s suggestion. However, due to the limited time for revision and the fact that it is time comsumable to breed the mice and collect the tissues, we will verify the levels of these genes with qPCR and WB in future.

Round 3

Reviewer 4 Report

The authors have considerably addressed all my points.

Reviewer 5 Report

accept